# Neural Conditional Transport Maps

**Carlos Rodriguez-Pardo**                                    *carlos.rodriguezpardo.jimenez@gmail.com*
*Department of Management, Economics and Industrial Engineering, Politecnico di Milano*
*RFF-CMCC European Institute on Economics and the Environment (EIEE)*
*Euro-Mediterranean Center on Climate Change (CMCC)*

**Leonardo Chiani**                                                    *leonardo.chiani@cmcc.it*
*Department of Management, Economics and Industrial Engineering, Politecnico di Milano*
*RFF-CMCC European Institute on Economics and the Environment (EIEE)*
*Euro-Mediterranean Center on Climate Change (CMCC)*

**Emanuele Borgonovo**                                            *emanuele.borgonovo2@gmail.com*
*Bocconi University, Department of Decision Sciences*
*MIT, Department of Nuclear Science and Engineering*

**Massimo Tavoni**                                                    *massimo.tavoni@cmcc.it*
*Department of Management, Economics and Industrial Engineering, Politecnico di Milano*
*RFF-CMCC European Institute on Economics and the Environment (EIEE)*
*Euro-Mediterranean Center on Climate Change (CMCC)*

## Abstract

We present a neural framework for learning conditional optimal transport (OT) maps between probability distributions. Conditional OT maps are transformations that adapt based on auxiliary variables, such as labels, time indices, or other parameters. They are essential for applications ranging from generative modeling to uncertainty quantification of black-box models. However, existing methods for generating conditional OT maps face significant limitations: input convex neural networks (ICNNs) impose severe architectural constraints that limit expressivity. At the same time, simpler conditioning strategies, such as concatenation, fail to model fundamentally different transport behaviors across conditions. Our approach introduces a conditioning mechanism capable of simultaneously processing both categorical and continuous conditioning variables, using learnable embeddings and positional encoding. At the core of our method lies a hypernetwork that generates transport layer parameters based on these inputs, creating adaptive mappings that outperform simpler conditioning methods. We showcase the framework's practical impact through applications to global sensitivity analysis, enabling efficient computation of OT-based sensitivity indices for complex black-box models. This work advances the state-of-the-art in conditional optimal transport, enabling broader application of optimal transport principles to complex, high-dimensional domains such as generative modeling, black-box model explainability, and scientific computing.

## 1   Introduction

Optimal transport (OT) is a powerful mathematical framework for comparing and transforming probability distributions, with wide applications across machine learning, computer vision, and scientific computing. OT provides a natural geometry for distribution spaces, offering stronger theoretical guarantees compared to alternative measures (Peyré & Cuturi, 2019). The importance of OT in machine learning has grown significantly, particularly in generative modeling where it forms the foundation of flow matching (Lipman et al.,

2023) and rectified flows (Liu et al., 2023). Despite its theoretical appeal and practical success, applying OT to high-dimensional, real-world problems has long been constrained by computational limitations. Classical OT methods scale poorly with sample size. Neural approaches have made significant progress in addressing these challenges by approximating transport maps with neural networks (Arjovsky et al., 2017; Makkuva et al., 2020; Korotin et al., 2023), enabling efficient computation in high-dimensional spaces.

The optimal mapping between distributions and the related cost are two elements of interest in the OT framework. The optimal mapping is used in different fields, ranging from computer vision (Bonneel & Digne, 2023) to biology (Bunne et al., 2023). However, many practical applications require *conditional* optimal maps: transformations that adapt based on auxiliary variables such as labels, time indices, or other parameters. Conditional transformations in general are crucial across diverse domains: generative models need to produce samples conditioned on class labels or continuous attributes, domain adaptation requires transport maps that vary with domain-specific characteristics, and climate-economy models need to model how distributions of climate variables change based on emissions or policy scenarios. The challenge lies in efficiently computing these conditional optimal maps, particularly in data-intensive problems where traditional OT methods become computationally prohibitive.

The OT cost has recently been presented as a valuable tool for global sensitivity analysis (GSA, Wiesel (2022); Borgonovo et al. (2024)). This application exemplifies the computational challenge for conditional OT. GSA leverages statistical methods to quantify how uncertainty in model outputs can be attributed to different input sources (Saltelli, 2002). GSA is therefore useful for understanding complex black-box models in machine learning, climate science, and economics (Saltelli et al., 2020). Recent works leverage OT costs to define sensitivity indices with valuable theoretical properties (Wiesel, 2022; Borgonovo et al., 2024). However, these methods face a severe computational bottleneck: computing sensitivity indices based on OT for complex models with many inputs requires solving hundreds to thousands of OT problems. The classical approach becomes computationally intractable due to memory requirements for storing cost matrices and the combinatorial explosion of required OT solvers. Efficiently learning conditional transport maps provides new tools for large-scale black-box model explainability.

Existing neural conditional OT methods suffer from fundamental limitations that prevent deployment in general cases (Bunne et al., 2022; Wang et al., 2025). The first approach is based on input convex neural networks (ICNNs), imposing severe architectural constraints, requiring non-negative weights, limited activation functions, and specialized initialization, leading to reduced expressivity and optimization difficulties, particularly for complex conditional problems (Korotin et al., 2021). Moreover, using concatenation as a conditioning strategy fails to model complex transport patterns across conditions. This type of conditioning lacks the flexibility to learn distinct mappings when different conditions require qualitatively different transformations, limiting its applicability to real-world scenarios where conditioning variables significantly alter the underlying transport structure. The second approach, based on conditional OT flows, alleviates some of these expressivity issues by conditioning the velocity field of an ODE on auxiliary variables, allowing the entire transport trajectory to depend on the condition. However, this comes at the cost of more complex training: solving ODE constraints requires additional regularization terms, hyperparameter tuning is delicate, and optimization is significantly more expensive, making it difficult to scale to large datasets or high-dimensional problems. Finally, Wang et al. (2025) restricts attention to simple tractable input distributions, while we consider the general setting where input distributions are arbitrary.

In this paper, we expand the state-of-the-art neural OT framework (Korotin et al., 2023) to efficiently learn conditional OT maps across both categorical and continuous conditioning variables. Our approach leverages a hypernetwork architecture—a neural network that dynamically generates the parameters for transport layers based on conditioning inputs. This mechanism creates adaptive mappings outperforming simpler conditioning methods while maintaining computational efficiency. Our contributions are as follows:

- We extend the Neural Optimal Transport (NOT) framework to conditional settings.

- We introduce a unified conditioning mechanism capable of simultaneously processing both categorical and continuous variables, using learnable embeddings and positional encoding.

- After testing a variety of possible alternatives, we propose a hypernetwork-based architecture that generates condition-specific transformation parameters, enabling fundamentally different mappings for each condition value without the computational overhead of separate networks.

- We provide empirical validation across synthetic datasets, image generation tasks, climate data, and integrated assessment models.

- We demonstrate practical applications to global sensitivity analysis, enabling efficient computation of OT-based sensitivity indices for complex black-box models at scale. This is the first comparison of the neural OT framework by Korotin et al. (2023) with well-established solvers for this task.

- We release our data and open-source implementation of our method in this Github repository.

The remainder of this paper is organized as follows. Section 2 reviews related work on traditional and neural OT, and conditioning methods. Section 3 details our conditional neural transport framework, including problem formulation, architecture, and training procedure. Section 4 presents the applications and experimental setup, while Section 5 shows results on benchmark datasets and ablations. Finally, Section 6 discusses limitations and future directions.

## 2 Background

### 2.1 Optimal Transport Foundations and Modern Applications

Optimal transport (OT) has emerged as a powerful mathematical framework across numerous domains, providing principled methods for comparing and transforming probability distributions. In machine learning, OT has been applied to generative modeling (Arjovsky et al., 2017), domain adaptation (Courty et al., 2017), and representation learning (Tolstikhin et al., 2018). Beyond traditional applications, OT has become foundational to modern generative modeling paradigms: flow matching (Lipman et al., 2023) uses optimal transport displacement interpolation to define probability paths for training continuous normalizing flows, while rectified flows (Liu et al., 2023) employ OT principles to learn straight transport paths for efficient generative modeling. In graphics and computer vision, OT enables geometry processing (Solomon et al., 2015), point cloud analysis (Bonneel et al., 2016), image color transfer (Ferradans et al., 2014), and texture synthesis (Gao et al., 2019).

The growing importance of OT in scientific computing is demonstrated by its application to GSA (Wiesel, 2022; Borgonovo et al., 2024), where OT-based sensitivity indices offer valuable theoretical properties for understanding complex black-box models. However, classical OT methods face severe computational limitations when applied to real-world problems, motivating the development of neural approaches that can scale to modern machine learning applications.

### 2.2 Neural Optimal Transport

Neural optimal transport methods were initially developed to address the computational bottlenecks of classical OT solvers. Early approaches include Wasserstein GANs (Arjovsky et al., 2017), which approximate Wasserstein distances through adversarial training but do not compute explicit transport maps. Later methods leveraged Brenier's theorem (Brenier, 1991) to implement Monge maps using input convex neural networks (ICNNs) (Amos et al., 2017; Makkuva et al., 2020).

However, ICNN-based approaches suffer from fundamental limitations that restrict their applicability. The convexity requirement imposes strict architectural constraints: all weights must be non-negative, activation functions are limited to convex choices (typically ReLU), and specialized initialization procedures are required to maintain convexity during training. More recently, Korotin et al. (2023) introduced Neural Optimal Transport (NOT) and Kernel NOT (Korotin et al., 2022), which bypass ICNN limitations through a minimax formulation that supports general cost functions and stochastic transport maps. NOT uses unconstrained neural networks for transport maps and critic functions, enabling flexible architectures while maintaining theoretical guarantees. We build upon this work for our conditional OT formulation.

## 2.3 Conditional Transport Challenges

**Computational Bottlenecks:** Traditional approaches to conditional transport problems require solving multiple independent OT problems for different conditioning values. This becomes computationally prohibitive at scale—for instance, global sensitivity analysis applications require computing OT-based sensitivity indices by solving *hundreds to thousands* of individual OT problems for each input variable and partition (Borgonovo et al., 2024). This approach is intractable for high-dimensional models with large datasets due to memory requirements for storing cost matrices and the combinatorial explosion of required solvers.

**Methodological Limitations:** Existing conditional OT methods suffer from architectural constraints that prevent effective scaling. CondOT (Bunne et al., 2022) relies on ICNNs, inheriting all the expressivity and optimization problems discussed above, which are further amplified in conditional settings where different conditions may require qualitatively different transport behaviors. Alternative approaches (Wang et al., 2023) impose architectural restrictions and assumptions on the input distributions that limit their flexibility. Furthermore, these methods typically use simple conditioning strategies like concatenation, which cannot model the different transport behaviors required when conditioning variables significantly alter the underlying distribution structure.

**Benchmarking Limitations:** Unlike other areas of machine learning, conditional optimal transport lacks standardized benchmarking and evaluation protocols. This absence hinders systematic comparison across different approaches and limits progress in the field. Existing methods often evaluate on different datasets with varying experimental setups, making it difficult to assess their relative strengths and weaknesses. Moreover, existing approaches focus on implementing the neural OT solver, while we also test various conditioning architectures.

## 2.4 Conditioning Mechanisms in Generative Models

The design of effective conditioning mechanisms is crucial for adaptive generative models. Conditioning approaches have evolved significantly, creating a spectrum of expressivity and computational complexity. Simple concatenation (Ho et al., 2020; Mirza & Osindero, 2014) directly combines feature vectors with conditioning information but lacks the flexibility to model condition-dependent behaviors. More sophisticated approaches include classifier guidance (Dhariwal & Nichol, 2021), which uses external classifiers to guide generation; normalization-based methods like FiLM (Perez et al., 2018) and AdaIN (Huang & Belongie, 2017), which modulate feature statistics; and attention mechanisms (Vaswani et al., 2017; Rombach et al., 2022), which allow selective focus on relevant conditioning information.

At the high end of the expressivity spectrum, hypernetworks (Ha et al., 2017) dynamically generate network parameters based on conditioning inputs, enabling fundamentally different computations for different conditions. This capability is particularly valuable for OT applications, where different conditioning values may require distinct mapping strategies that cannot be achieved through feature modulation alone.

For conditional transport maps specifically, the choice of conditioning mechanism is critical. Simple concatenation, as used in CondOT (Bunne et al., 2022), fails when different conditions require qualitatively different transport behaviors—for example, when transporting between fundamentally different distribution types or when conditioning variables induce non-linear changes in the optimal transport structure. Our hypernetwork-based approach addresses this limitation by generating condition-specific transformation parameters, providing the expressiveness of separate networks per condition while maintaining computational efficiency through parameter sharing in the hypernetwork itself.

This work provides the first systematic comparison of conditioning mechanisms within the neural optimal transport framework, demonstrating the superior performance of hypernetwork-based approaches for learning flexible conditional transport maps.

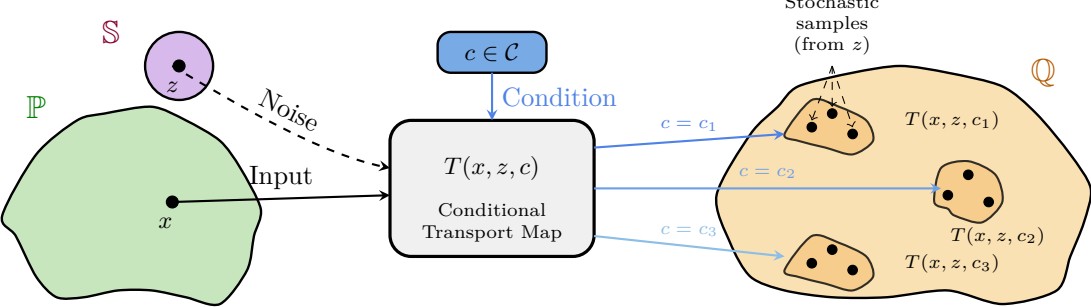

Figure 1: A diagram of our conditional OT map framework. Our transport network $T$ takes as input samples $x \sim \mathbb{P}$ and transports them to $\mathbb{Q}$, conditioned by $c$. Optionally, $T$ can receive additional noise inputs $z \sim \mathbb{S}$, from a known probability distribution $\mathbb{S}$, introducing stochasticity.

## 3  Neural Conditional Transport Maps

Our solution extends the Neural Optimal Transport (NOT) framework of Korotin et al. (2023) with a flexible conditioning architecture that enables efficient learning of conditional transport maps across both categorical and continuous conditioning variables. At the core of our method lies a hypernetwork that dynamically generates transport layer parameters based on conditioning inputs, providing high expressivity while maintaining computational efficiency through shared parameter generation. This design allows different conditions to induce different mapping strategies—essential when conditional distributions require distinct transport behaviors that cannot be captured through simple feature concatenation or modulation approaches.

Our framework offers several key advantages over existing methods: (1) it avoids the architectural constraints of ICNN approaches, enabling flexible network designs with standard optimization procedures; (2) it scales efficiently to large numbers of conditioning values without requiring separate OT solvers, addressing the computational bottlenecks in sensitivity analysis applications; (3) it provides a unified mechanism for handling both discrete and continuous conditioning variables through learnable embeddings and positional encoding; and (4) it enables systematic comparison of conditioning strategies within the same neural OT formulation.

We illustrate our conditional OT maps in Figure 1. In this section, we begin by formalizing the conditional OT problem (Section 3.1), then describe our architecture that leverages encoder-decoder structures for both transport map $T$ and critic function $f$ (Section 3.2). We then introduce our conditioning mechanism that handles discrete variables and continuous partitions (Section 3.3), present architectural variants including our hypernetwork approach and alternative conditioning strategies (Section 3.4), and detail our training procedure with a custom pretraining strategy that addresses the optimization challenges common in conditional transport learning (Section 3.5). Implementation details are provided in the supplementary material.

### 3.1  Problem formulation

To address the computational bottlenecks in conditional OT applications, we formulate the problem following the neural OT framework that enables efficient scaling to large numbers of conditioning values without requiring separate solvers for each condition. Let us consider two probability measures $\mu$ and $\nu$ defined over two Polish spaces, $\mathcal{X} \subset \mathbb{R}^m$ and $\mathcal{Y} \subset \mathbb{R}^n$, respectively. Let's also define a lower-semicontinuous cost function $k \colon \mathcal{X} \times \mathcal{Y} \longrightarrow [0, +\infty]$ such that $k(y, y') = 0$ if and only if $y = y'$. The Kantorovich formulation of the OT problem can be stated as follows:

$$K(\mu, \nu) = \inf_{\pi \in \Pi(\mu, \nu)} \int_{\mathcal{X} \times \mathcal{Y}} k(x, y)\, d\pi(x, y), \tag{1}$$

where $\Pi(\mu, \nu)$ is the set of joint probability measures on $\mathcal{X} \times \mathcal{Y}$ with marginals $\mu$ and $\nu$.

Even though the primal formulation of the OT problem in equation 1 is easier to interpret, the dual formulation is more useful from a practical perspective, especially using neural networks. First, we introduce the concept

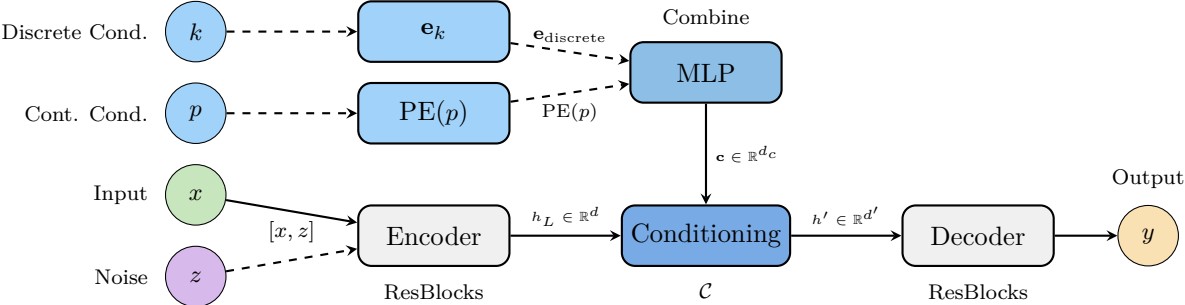

Figure 2: Architecture of the conditional networks (transport $T$ and critic $f$). The encoder processes input data ($[x, z]$ for $T$ or $x$ for $f$) through residual blocks to produce latents $h_L \in \mathbb{R}^d$. We combine discrete variable embeddings $\mathbf{e}_k$ with positional encoding of continuous values $PE(p)$ through an MLP to produce the unified conditioning vector $\mathbf{c} \in \mathbb{R}^{d_c}$. The conditioning module $\mathcal{C}$ transforms this latent into $h' \in \mathbb{R}^{d'}$, which the decoder processes into the final output. The noise $z$ is only used in $T$.

of $k$-transform. The $k$-transform $f^k$ of a function $f : \mathcal{Y} \longrightarrow \mathbb{R}$ is defined as $f^k(x) := \inf_{y \in \mathcal{Y}} \{k(x, y) - f(y)\}$. The dual formulation of the Kantorovich OT problem is then:

$$K(\mu, \nu) = \sup_f \left[ \int_{\mathcal{X}} f^k(x) d\mu(x) + \int_{\mathcal{Y}} f(y) d\nu(y) \right]. \tag{2}$$

For our purposes, the dual problem can be further reformulated as a maximin problem (Korotin et al., 2023). We introduce a third atomless distribution, $\omega$, defined over the Polish space $\mathcal{Z} \subset \mathbb{R}^s$. Given a measurable map $T : \mathcal{X} \times \mathcal{Z} \longrightarrow \mathcal{Y}$ and its push-forward operator $T\#$, the maximin formulation is:

$$K(\mu, \nu) = \sup_f \inf_T \mathcal{L}(f, T), \tag{3}$$

where

$$\mathcal{L}(f, T) = \int_{\mathcal{Y}} f(y) \, d\nu(y) + \int_{\mathcal{X}} \left( k(x, T(x, \cdot)\#\omega) - \int_{\mathcal{Z}} f(T(x, z)) \, d\omega(z) \right) d\mu(x). \tag{4}$$

This formulation allows our neural approach to amortize computation across conditioning values, avoiding the prohibitive cost of solving separate OT problems for each condition. We note here that the results in Korotin et al. (2023) can be extended to *weak* costs, but they are out of the scope of this work.

**The conditioned problem.** We start from equation 4 to integrate the conditioning mechanism into the problem formulation, enabling applications like global sensitivity analysis to compute transport-based indices efficiently across multiple conditioning values with a single learned model. We assume that the condition $c$ belongs to a measure space $\mathcal{C}$. From the theoretical perspective, the extension is straightforward:

$$K(\mu, \nu, c) = \sup_f \inf_T \mathcal{L}(f, T, c), \tag{5}$$

where

$$\mathcal{L}(f, T, c) = \int_{\mathcal{Y}} f(y, c) \, d\nu(y) + \int_{\mathcal{X}} \left( k(x, T(x, \cdot, c)\#\omega) - \int_{\mathcal{Z}} f(T(x, z, c), c) \, d\omega(z) \right) d\mu(x). \tag{6}$$

### 3.2 Neural Optimal Transport

Building on the NOT framework of Korotin et al. (2023), we parametrize both the transport map $T$ and critic function $f$ using neural networks. We employ encoder-decoder architectures that enable flexible conditioning in the latent space while supporting various layer types (linear, convolutional, attention) based on the data modality. We illustrate our model design in Figure 2.

**Transport & Critic:** The transport map $T : \mathcal{X} \times \mathcal{Z} \times \mathcal{C} \to \mathcal{Y}$ is implemented as a neural network with encoder-decoder structure. For the common case where $\mathcal{X} = \mathcal{Y} = \mathbb{R}^n$, the encoder maps $\mathbb{R}^{n+s}$ into $\mathbb{R}^d$, a conditioning

---

**Algorithm 1** Training Neural Conditional Optimal Transport

---

1: **Input:** Distributions $\mathbb{P}$, $\mathbb{Q}$, $\mathbb{S}$ accessible by samples; Transport network $T_\theta \colon \mathbb{R}^P \times \mathbb{R}^S \times \mathcal{C} \to \mathbb{R}^Q$; Critic network $f_\omega \colon \mathbb{R}^Q \times \mathcal{C} \to \mathbb{R}$; Cost $\mathcal{L} \colon \mathcal{X} \times \mathcal{Y} \to \mathbb{R}$; Conditioning distribution $\mathcal{C}$; Maximum steps $N$, Transport iterations per step $K_T$
2: **Pre-training:** Initialize $T_\theta$ and $f_\omega$ using objectives in Eq. (3)-(7)
3: **Output:** Learned conditional transport map $T_\theta$
4: **for** $t = 1, 2, \ldots, N$ **do**
5:     Sample conditioning $c \sim \mathcal{C}$
6:     Sample batches $Y \sim \mathbb{Q}$, $X \sim \mathbb{P}$; for each $x \in X$ sample $Z_x \sim \mathbb{S}$
7:     $\mathcal{L}_f \leftarrow \frac{1}{|X|} \sum_{x \in X} \frac{1}{|Z_x|} \sum_{z \in Z_x} f_\omega(T_\theta(x, z, c), c) - \frac{1}{|Y|} \sum_{y \in Y} f_\omega(y, c)$
8:     Update $\omega$ using $\frac{\partial \mathcal{L}_f}{\partial \omega}$ (gradient ascent)
9:     **for** $k_T = 1, 2, \ldots, K_T$ **do**
10:         Sample conditioning $c \sim \mathcal{C}$
11:         Sample batch $\tilde{X} \sim \mathbb{P}$; for each $x \in \tilde{X}$ sample $\tilde{Z}_x \sim \mathbb{S}$
12:         $\mathcal{L}_T \leftarrow \frac{1}{|\tilde{X}|} \sum_{x \in \tilde{X}} \left[ \mathcal{L}(x, T_\theta(x, \tilde{Z}_x, c)) - \frac{1}{|\tilde{Z}_x|} \sum_{z \in \tilde{Z}_x} f_\omega(T_\theta(x, z, c), c) \right]$
13:         Update $\theta$ using $\frac{\partial \mathcal{L}_T}{\partial \theta}$
14:     **end for**
15: **end for** =0

---

module transforms $\mathbb{R}^d \times \mathcal{C}$ into $\mathbb{R}^{d'}$, and the decoder maps $\mathbb{R}^{d'}$ into $\mathbb{R}^n$. The complete transport map is: $T(x, z, c) = \mathrm{Decoder}_T(\mathrm{Conditioning}_T(\mathrm{Encoder}_T([x, z]), c))$ where $[x, z] \in \mathbb{R}^{n+s}$ denotes the concatenation of input $x$ and noise $z$, $d$ is the latent dimension, and $d'$ is the conditioned latent dimension. The critic function $f \colon \mathcal{Y} \times \mathcal{C} \to \mathbb{R}$ follows the same structure without noise input: $f(y, c) = \mathrm{Decoder}_f(\mathrm{Conditioning}_f(\mathrm{Encoder}_f(y), c))$

**Layer Design:** Unlike CondOT (Bunne et al., 2022), which uses ICNNs, our unconstrained architecture supports flexible designs with standard optimization procedures. We use residual blocks He et al. (2016) for both $T$ and $f$, along with layer normalization Ba et al. (2016) and orthogonal initialization Saxe et al. (2014). Each block consists of $\mathrm{ResBlock}(h) = h + \alpha \cdot \mathcal{F}(h)$, where $\alpha$ is a learnable scaling parameter and $\mathcal{F}$ represents a composite function that may include normalization, non-linear activations, and appropriate transformations for the data type.

**Encoder-Decoder Design:** The encoder produces a sequence of hidden states $\{h_1, h_2, ..., h_L\}$ where $h_L \in \mathbb{R}^d$ is used for conditioning. The decoder takes the conditioned representation and maps to the output space. Both modules can be instantiated with different layer types depending on the data modality: $\mathrm{Encoder}(x) = h_L$ where $h_i = \mathrm{Layer}_i(h_{i-1})$ and $h_0 = x$. Here, $\mathrm{Layer}_i$ can be any differentiable layer. The encoder learns condition-invariant features from the source distribution $\mathbb{P}$, while the decoder applies condition-specific transformations. Importantly, this flexibility allows $\mathrm{Layer}_i$ to include any standard neural network components.

**Latent Space Conditioning:** The conditioning operates on the latent representation $h_L \in \mathbb{R}^d$ from the encoders. Given a condition $c \in \mathcal{C}$, the conditioning function transforms the latent representation before passing it to the decoder. This design allows the network to learn condition-specific transformations while sharing feature extraction across conditions.

### 3.3 Conditioning mechanism

Our framework supports conditioning on discrete and continuous variables, with the capability to enable either or both types of conditioning depending on the application. The conditioning module transforms these inputs into a unified representation that modulates the transport map, enabling efficient scaling to applications requiring transport across many conditioning values without solving separate OT problems.

For **discrete variables** (e.g., categorical features with $K$ possible values), we use learnable embeddings. Each variable $k \in \{0, 1, ..., K-1\}$ is mapped to an embedding $\mathbf{e}_k \in \mathbb{R}^{d_c}$, where $d_c$ is the condition dimensionality, defined as $\mathbf{e}_k = \mathcal{E}[k]$ with $\mathcal{E} \in \mathbb{R}^{K \times d_c}$. For **continuous variables**, based on Transformers Vaswani et al. (2017) and Radiance Fields Mildenhall et al. (2020), we use sinusoidal positional encodings to preserve the continuous

nature while providing a rich representation: $\text{PE}(p, 2i) = \sin\left(\frac{p}{10000^{2i/d}}\right)$, $\text{PE}(p, 2i+1) = \cos\left(\frac{p}{10000^{2i/d}}\right)$, where $p$ is the min-max normalized continuous value, $d$ is the encoding dimension, and $i \in \{0, 1, ..., d/2 - 1\}$. Additionally, we support other encoding strategies including Fourier features, learned embeddings, or scalar values, allowing flexible adaptation to different problem domains.

**Unified Conditioning.** The conditioning module flexibly handles discrete variables, continuous variables, or both. When both types are present, their representations are concatenated, then processed to produce the final conditioning vector $\mathbf{c} = \text{MLP}([\mathbf{e}_{\text{discrete}}, \text{PE}(p_{\text{continuous}})])$, where $[\cdot, \cdot]$ denotes concatenation and MLP is a multi-layer perceptron with SiLU Ramachandran et al. (2017) activations. The resulting vector $\mathbf{c} \in \mathbb{R}^{d_c}$ is used to modulate the transport map in the latent space.

**Flexible Configuration.** Both discrete and continuous conditioning are optional and can be independently enabled or disabled. The module requires at least one type of conditioning to be active. This flexibility allows our framework to adapt to various application domains—from purely categorical problems to continuous spatiotemporal transport tasks. Unlike CondOT (Bunne et al., 2022), we use different conditioning embeddings for T and f, as this showed better performance. This separation proves to be quantitatively important, suggesting that the transport map and critic function use conditioning information in fundamentally different ways—the transport map must adapt its mapping strategy based on conditions, while the critic must evaluate transport quality conditional on the same information.

### 3.4 Conditioning modules variants

We explored several designs for applying the conditioning $\mathbf{c} \in \mathbb{R}^{d_c}$ to modulate the transport map.

**Hypernetwork Conditioning** The hypernetwork generates the final layer weights and biases of the encoder based on the conditioning. Given the encoder output $\mathbf{h} \in \mathbb{R}^d$ and conditioning $\mathbf{c}$, the hypernetwork $\mathcal{H}$ is a shallow MLP that generates parameters: $[\mathbf{W}, \mathbf{b}] = \mathcal{H}(\mathbf{c})$, where $\mathbf{W} \in \mathbb{R}^{d \times n}, \mathbf{b} \in \mathbb{R}^n$. The final output is then computed as: $\mathbf{y} = \mathbf{h}\mathbf{W} + \mathbf{b}$. Unlike feature modulation approaches that can only adjust existing computations, generating weights enables fundamentally different transformations per condition—addressing the core limitation of simple conditioning strategies that fail when different conditions require qualitatively distinct transport behaviors. This approach provides significant expressiveness while maintaining the computational efficiency of a single shared architecture, directly addressing the scalability bottlenecks in applications like global sensitivity analysis.

**Alternative Conditioning Mechanisms** We evaluated several conditioning strategies beyond our hypernetwork approach. The simplest baseline is *Concatenation*, which directly combines feature vectors with the condition but cannot model the fundamentally different transport behaviors required when conditions significantly alter the optimal mapping strategy. More sophisticated approaches include *Cross-Attention* (Vaswani et al., 2017), which uses attention mechanisms to modulate features, and *Feature-wise Linear Modulation (FiLM)* (Perez et al., 2018), which applies learnable transformations to existing features. However, these methods are fundamentally limited to adjusting or re-weighting existing computations rather than enabling entirely different computational paths per condition. We also tested normalization-based methods like *Adaptive Instance Normalization* (Huang & Belongie, 2017) and *Conditional Layer Normalization* (Su et al., 2021), along with attention-inspired techniques such as *Squeeze-and-Excite* (Hu et al., 2018) and *Feature-wise Affine Normalization (FAN)* (Zhou et al., 2021), but these approaches similarly cannot generate the qualitatively different transport behaviors required when conditions induce fundamentally different optimal mappings. Although these alternatives showed promise in specific scenarios, our hypernetwork approach consistently demonstrated superior performance across all benchmarks. Notably, our lightweight hypernetwork implementation proved to be computationally efficient in training time, making it a sound choice even for low-budget scenarios. We ablate these components on Section 5.

### 3.5 Training Procedure

During training, the transport map $T$ and critic function $f$ must maintain adversarial balance while adapting to diverse conditioning values. This creates optimization challenges, where networks must simultaneously

learn transport mappings and adapt to diverse conditioning values. We address them through a two-phase approach: pre-training for stable initialization followed by minimax optimization.

**Pre-training:** Randomly initialized networks may implement highly non-linear transformations far from identity, leading to unstable optimization—a problem that is particularly acute in conditional transport learning where networks must balance transport objectives with condition-dependent adaptation. Motivated by this observation, we introduce a lightweight pre-training that establishes favorable initial conditions for both networks. We pre-train $T$ to approximate the identity mapping: $\mathcal{L}_T^{\text{pre}} = \mathbb{E}_{x\sim\mathbb{P},z\sim\mathbb{S},c\sim\mathcal{C}}[\|T(x,z,c) - x\|_2^2]$. This is important in conditional settings because the transport map must learn to interpolate between potentially very different optimal mappings for different conditioning values—starting from identity provides a stable baseline that prevents the network from collapsing to degenerate solutions that work for some conditions but fail for others. This initialization ensures that, early in training, the transport map preserves input structure. Besides, we pre-train $f$ with a multi-objective loss: $\mathcal{L}_f^{\text{pre}} = \lambda_{\text{smooth}}\mathcal{L}_{\text{smooth}} + \lambda_{\text{transport}}\mathcal{L}_{\text{transport}} + \lambda_{\text{mag}}\mathcal{L}_{\text{mag}}$. First, the smoothness term prevents sharp discontinuities that lead to gradient explosion during adversarial training:$\mathcal{L}_{\text{smooth}} = \mathbb{E}_{x\sim\mathbb{P},c\sim\mathcal{C}}[\|f(x+\epsilon,c) - f(x,c)\|_2^2]$ where $\epsilon \sim \mathcal{N}(0,\sigma^2)$. In our formulation, $f$ must evaluate transport quality consistently across diverse conditioning values, making smoothness important to prevent the adversarial training from destabilizing when conditions require very different transport behaviors. Second, the transport term maintains the OT objective: $\mathcal{L}_{\text{transport}} = \mathbb{E}_{x\sim\mathbb{P},z\sim\mathbb{S},c\sim\mathcal{C}}[f(T(x,z,c),c)] - \mathbb{E}_{y\sim\mathbb{Q},c\sim\mathcal{C}}[f(y,c)]$. This helps $f$ learn meaningful discrimination between transported and target samples from initialization. Finally, a magnitude control term prevents unbounded growth, enhancing stability: $\mathcal{L}_{\text{mag}} = \mathbb{E}_{y\sim\mathbb{Q},c\sim\mathcal{C}}[(|f(y,c)|-1)^2]$.

**Optimization:** Following pre-training, we employ an alternating training that reflects the minimax structure of optimal transport. We detail our approach in Algorithm 1. Following Korotin et al. (2023), we perform $K_T > 1$ updates for the transport map per critic update. This asymmetry addresses the imbalance in learning complexity—the transport map must learn condition-dependent transformations while maintaining transport constraints, whereas the critic primarily serves as a measure of transport quality. We find $K_T \in [4,6]$ provides an adequate balance between training stability and efficiency.This asymmetric training schedule proves essential for conditional transport learning, where the complexity imbalance between transport and critic functions is amplified by the need to handle diverse conditioning values.

**Conditions Sampling:** The performance of this learning procedure depends on how we sample from the conditioning space $\mathcal{C}$ during training. For discrete variables, we use uniform sampling across all categories. For continuous variables, we employ Beta distribution sampling $\text{Beta}(\alpha,\beta)$ with symmetric parameters ($\alpha = \beta = 0.95$), which slightly oversamples values around the min and max values in the training datasets. This strategy prevents underfitting in boundary conditions. This sampling strategy is important for applications like global sensitivity analysis, where transport quality must be maintained across the full range of conditioning values to ensure accurate sensitivity indices.

## 4 Applications and Experimental Setup

We evaluate our conditional neural transport framework across diverse applications that span scientific computing, sensitivity analysis, and image generation. These applications are specifically chosen to demonstrate the breadth of conditioning scenarios our framework can handle: from purely discrete conditioning to hybrid discrete-continuous conditioning, and from lower-dimensional scientific problems to image generation tasks. Each application presents unique challenges that test different aspects of our conditional transport approach, providing comprehensive validation of the framework's capabilities.

### 4.1 Climate Economic Impact Distribution Transport

We examine the economic impacts of climate change using the empirical damage function model of Burke et al. (2015). This application requires building an emulator for complex multivariate distributions conditioned on both categorical (scenario) and continuous (time) variables, representing a challenging test case for hybrid conditioning mechanisms.

**Problem Formulation.** Let $\mathcal{X} \subset \mathbb{R}^n$ represent the space of GDP per capita with climate damages across $n = 20$ countries. For each country $i$, the impact is quantified through 1000 bootstrap replicates, resulting in

a high-dimensional empirical distribution that captures the uncertainty in damage estimates. We define our conditioning space $\mathcal{C} = \mathcal{C}_{\text{ssp}} \times \mathcal{C}_{\text{year}}$, where $\mathcal{C}_{\text{ssp}} = \{0, 1, 2, 3\}$ represents four SSP scenarios (SSP1-1.9, SSP2-4.5, SSP3-7.0, SSP5-8.5) and $\mathcal{C}_{\text{year}} = [2030, 2100]$ represents projection years. Our goal is to learn a conditional transport map $T_\theta : \mathbb{R}^n \times \mathcal{Z} \times \mathcal{C} \to \mathcal{X}$ that efficiently transforms samples from a reference distribution to match target distributions under different climate scenarios and future years.

**Dataset and Evaluation.** The dataset contains empirical damage distributions for 20 countries across 4 SSP scenarios and 71 years (2030-2100), totaling 5,680 conditioning combinations. Each distribution is characterized by 1000 bootstrap samples, providing rich uncertainty quantification. We use 80% of the data for training and 20% for evaluation, ensuring temporal and scenario stratification. We evaluate performance using the Wasserstein-2 distance between generated and target distributions-

## 4.2 Global Sensitivity Analysis for Integrated Assessment Models

Our second application focuses on global sensitivity analysis (GSA) for the RICE50+ Integrated Assessment Model (IAM) (Gazzotti, 2022), using the optimal transport-based sensitivity indices presented in Borgonovo et al. (2024). This application demonstrates our framework's ability to replace computationally intensive traditional approaches with efficient neural approximations.

**Theoretical Background.** Let $\mathbf{Y} = (Y_1, \ldots, Y_k)$ represent quantities of interest as a function of inputs $\mathbf{C} = (C_1, \ldots, C_d)$. Assume $\mathbf{C}$ and $\mathbf{Y}$ are random vectors on probability space $(\Omega, \mathcal{B}, \mu)$, with $\mu_{C_i}$ and $\mu_{\mathbf{Y}}$ denoting the distributions of $C_i$ and $\mathbf{Y}$, respectively. Consider a model $\mathbf{f} : \mathcal{C} \subset \mathbb{R}^d \to \mathbb{R}^k$. The OT-based sensitivity index for the $i$-th input is:

$$\iota^K(\mathbf{Y}, C_i) = \frac{\mathbb{E}_{C_i}[K(\mu_{\mathbf{Y}}, \mu_{\mathbf{Y}|C_i})]}{\mathbb{E}[k(\mathbf{Y}, \mathbf{Y}')]}, \tag{7}$$

where $K(\mu_{\mathbf{Y}}, \mu_{\mathbf{Y}|C_i})$ is the OT cost between unconditional and conditional output distributions, and the denominator provides normalization. This index satisfies zero-independence ($\iota^K(\mathbf{Y}, C_i) = 0$ iff $\mathbf{Y}$ independent of $C_i$) and max-functionality ($\iota^K(\mathbf{Y}, C_i) = 1$ iff $\mathbf{Y} = \mathbf{g}(C_i)$ for some function $\mathbf{g}$).

**Problem Formulation.** Let $f : \mathcal{C} \subset \mathbb{R}^3 \to \mathcal{Y} \subset \mathbb{R}^{58}$ be the RICE50+ model mapping input parameters $\mathbf{C} \in \mathcal{C}$ to output $\mathbf{Y} \in \mathcal{Y}$. RICE50+ (Gazzotti, 2022) is an IAM with high regional heterogeneity used to assess climate policy benefits and costs. We estimate the sensitivity of $CO_2$ emissions for a single region to three inputs related to emissions abatement costs (`klogistic`), aversion to inter-country inequality (`gamma`), and climate impacts (`kw_2`), leveraging data from Chiani et al. (2025).

Traditional computation of these indices requires solving multiple OT problems for each conditioning variable and partition—specifically, for $d$ variables with $M$ partitions each, this requires $d \times M$ separate OT solutions, becoming computationally intractable for large-scale applications. We partition each input space into $M = 25$ bins and define conditioning space $\tilde{\mathcal{C}} = \{0, 1, 2\} \times [0, 1]$, where the first component represents variable index and the second represents normalized partition. Our approach learns a single conditional transport map $T_\theta : \mathcal{Y} \times \mathcal{Z} \times \tilde{\mathcal{C}} \to \mathcal{Y}$ that can compute sensitivity indices for all variables and partitions efficiently.

**Dataset and Evaluation.** The dataset contains 10,000 model evaluations with corresponding input-output pairs. We use the traditional estimator with partitioned inputs as ground truth, computed using transportation simplex solvers. We evaluate our model with the Pearson correlation $\rho$ between neural and simplex-computed costs.

## 4.3 Image Generation Tasks

To demonstrate the generality of our approach beyond scientific computing, we evaluate conditional image generation on standard computer vision benchmarks. These tasks test our framework's ability to handle high-dimensional visual data with both discrete class labels and continuous attribute conditioning.

**MNIST Conditional Generation.** We perform conditional digit generation with discrete conditioning, using class labels $\mathcal{C} = \{0, \ldots, 9\}$. For this application, we use convolutional models for both $T$ and $f$, which

include convolutional layers with different values for strides and kernel sizes, therefore leveraging standard components for image generation models without imposing convexity constraints.

**Problem Formulation.** We learn a transport map $T_\theta : \mathcal{Z} \times \mathcal{C} \to \mathbb{R}^{28 \times 28}$ that transforms noise into digit images conditioned on class and attributes.

**Dataset and Evaluation.** We use the standard MNIST dataset augmented with rotation and stroke thickness variations to create ground truth conditional distributions. To evaluate this setup, we use the Fréchet Inception Distance (FID) for generation quality, and digit classification accuracy.

### 4.4 Experimental Protocols

**Hardware and Implementation.** All experiments are conducted on RTX 4070 GPU with i7-13700H CPU. Our framework is implemented in PyTorch with automatic mixed precision training. We use CodeCarbon (Lacoste et al., 2019) to monitor energy consumption across experiments.

**Architecture Details.** For climate and GSA applications, we use fully connected encoder-decoder architectures with 4 encoder layers (hidden size 128) and 8 decoder layers for transport networks, and 3 encoder/decoder layers for critic networks. For image generation, we employ convolutional architectures with residual blocks. All networks use layer normalization, orthogonal initialization, and learnable residual scaling parameters $\alpha$.

**Training Configuration.** We employ our two-phase training procedure: 500 steps of pre-training followed by alternating minimax optimization. Transport networks receive $K_T = 5$ updates per critic update. Learning rates are $2 \times 10^{-5}$ for transport and $3 \times 10^{-5}$ for critic networks, with AdamW optimization and weight decay of 0.03. Conditioning sampling uses uniform distribution for discrete variables and Beta$(0.95, 0.95)$ for continuous variables.

**Experimental Comparisons.** Our evaluation focuses on systematic ablation studies to identify optimal design choices within our framework. We compare different conditioning mechanisms including concatenation, cross-attention, Feature-wise Linear Modulation (FiLM), normalization-based approaches (Adaptive Instance Normalization, Conditional Layer Normalization), and attention-inspired techniques (Squeeze-and-Excite, Feature-wise Affine Normalization). We also evaluate different continuous variable encoding strategies (positional encoding, Fourier features, scalar values), architectural choices (shared vs. separate conditioning embeddings), and training procedures (with and without pretraining). For GSA applications, we validate against traditional transportation simplex solvers to establish ground truth sensitivity indices.

**Comparisons with CondOT** Bunne et al. (2022). We additionally compare against two CondOT baselines. *CondOT default* uses the original CondOT architecture and optimization recipe, together with a tokenized factorized conditioning scheme in which categorical and discretized continuous condition components are represented as separate conditioning tokens. Since the original CondOT formulation is not designed for mixed categorical: continuous conditioning in the form considered here, continuous variables (time indices or partition levels) must be discretized before being provided to the model. To provide a more informative comparison, we also define a *CondOT adapted* variant, in which we replace this discretized conditioning with a mixed conditioning mechanism combining one-hot categorical inputs and normalized continuous scalars. We do not include CondOT in the MNIST experiments, since its original formulation is not designed for convolutional architectures. For both configurations, we create architectures with a comparable number of trainable parameters to our models, to fully isolate the impact of architecture design and optimization choices rather than model capacity. Number of training epochs remain equal to our models, while optimization choices (optimizer betas, number of inner iterations) for CondOT follow those of their original implementation.

## 5 Results and Analysis

We present a comprehensive empirical evaluation of our conditional neural transport framework across the applications described in Section 4. Our evaluation follows a systematic approach: we first establish the effectiveness of our architectural improvements through ablation studies on both unconditional and conditional

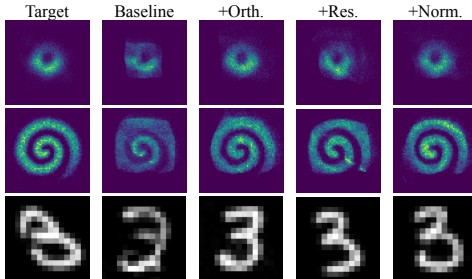

| Dataset | Metric | Baseline | +Orth. | +Res | +Norm |
|---|---|---|---|---|---|
| Black Hole | KL $\downarrow$ | 2.333 | 1.888 | 2.121 | 2.144 |
| | Wass. $\downarrow$ | 0.062 | 0.047 | 0.041 | 0.031 |
| Swiss Roll | KL $\downarrow$ | 1.081 | 0.904 | 0.879 | 0.813 |
| | Wass. $\downarrow$ | 0.089 | 0.052 | 0.041 | 0.015 |
| MNIST | MMD $\downarrow$ | 0.047 | 0.033 | 0.027 | 0.028 |
| | Wass. $\downarrow$ | 0.042 | 0.037 | 0.015 | 0.011 |
| | FID $\downarrow$ | 2.378 | 2.333 | 2.070 | 2.075 |
| | Acc. $\uparrow$ | 82.15 | 100 | 100 | 100 |

Figure 3: Results of our ablation study comparing different model configurations on an **unconditional** transport setting. The table presents quantitative metrics, while the right panel shows the visualization.

settings, then demonstrate the practical impact of our approach on real-world applications. Finally, we analyze computational aspects and discuss the implications of our findings.

## 5.1 Ablation Studies

To validate our design choices and quantify the contribution of each component, we conduct systematic ablation studies examining architectural improvements and conditional-specific design decisions.

### 5.1.1 Architectural Foundation

We first evaluate architectural improvements to the base NOT framework Korotin et al. (2023), establishing a strong foundation for our conditional extensions. Building upon their ReLU-based architectures, we incrementally add orthogonal initialization, residual connections, and layer normalization, while preserving identical training configuration and comparable parameter counts across models.

Figure 3 presents results across three datasets that test different aspects of neural optimal transport. The first two rows show 2D probability distributions using simple MLPs as baseline architecture. We measure KL divergence and Wasserstein loss between target and transported distributions (from a 2D Uniform prior), using $2^{15}$ samples. The third row showcases a convolutional model performing image-to-Image generation on MNIST (digit 2 to digit 3), evaluated with MMD distance, Wasserstein loss, FID score, and classification accuracy.

In both MLP and convolutional settings, our enhancements yield notable quantitative and qualitative improvements, as shown in the visualization panels. These architectural enhancements form the foundation of our conditional transport experiments, demonstrating that improved convergence can be achieved without increasing computational cost.

### 5.1.2 Conditional Framework Design

Having established our architectural foundation, we turn to the key design decisions specific to conditional transport learning. Table 1 presents our systematic evaluation of conditioning mechanisms, encoding strategies, and training procedures.

We compare our final configuration—using hypernetwork conditioning, pretraining, positional encoding, and separate conditioning embeddings for transport and critic networks—with systematic variations. Training times and parameter counts provide computational cost analysis, while application-specific metrics evaluate transport quality. For the climate damages dataset, we report Wasserstein distance and efficiency (Wasserstein $\times$ time). For IAM data, we report Pearson correlation $\rho$ between neural and simplex-computed costs and efficiency-adjusted correlation. All experiments use identical training protocols on RTX 4070 GPU with i7-13700H CPU, measuring 0.009323 kWh electricity consumption via CodeCarbon for the climate damages application Lacoste et al. (2019).

Table 1: Ablation study of our **conditional** transport framework. We report computational cost and accuracy on our datasets. Our final model (leftmost column) uses a hypernetwork with positional encoding and pretraining, without shared embedding. Each other column group represents variations from this configuration. We present best results for our method in **green bold**, second best are underlined, worst are red. Note that for the *MNIST* experiment, there is no continuous input variable, therefore we don't ablate their encodings. *CondOT default* uses the original CondOT architecture and optimization recipe with tokenized factorized conditioning over categorical and discretized continuous variables. *CondOT adapted* keeps the CondOT training recipe unchanged but replaces this discretized conditioning with mixed conditioning combining one-hot categorical inputs and normalized continuous scalars. CondOT is evaluated only on the climate and GSA/RICE tasks; MNIST entries are marked as not applicable.

| | CondOT(2022) | | Ours | Pretrain | Embed. | Conditioning Type | | | | | | | Continuous Encoding | |
| --- | --- | --- | --- | --- | --- | --- | --- | --- | --- | --- | --- | --- | --- | --- |
| | Default | Adapted | | False | Shared | Concat | SE | FiLM | FAN | Attn. | CLN | AdaIN | Scalar | Fourier |
| *Comp. cost* | | | | | | | | | | | | | | |
| Time (s) ↓ | 586 | 721 | 466 | **451** | 457 | **451** | 554 | 854 | 487 | 921 | 712 | 658 | 464 | 486 |
| Parameters (k) ↓ | 1216 | 1316 | 1158 | 1158 | 1156 | 1324 | 2764 | 2874 | 2861 | 2908 | 2791 | 2790 | **1150** | 1152 |
| *Climate Damages* | | | | | | | | | | | | | | |
| Wass. ↓ | 0.322 | 0.255 | **0.170** | 0.214 | 0.265 | 0.267 | 0.716 | 0.974 | 0.458 | 0.721 | 0.691 | 0.744 | 0.255 | 0.172 |
| Wass. × time ↓ | 188.7 | 183.8 | **79.22** | 96.51 | 121.1 | 120.4 | 396.7 | 831.8 | 223.0 | 664.0 | 491.9 | 489.6 | 118.3 | 83.59 |
| *IAM Data* | | | | | | | | | | | | | | |
| ρ ↑ | 0.025 | 0.389 | **0.942** | 0.905 | 0.812 | 0.305 | 0.377 | 0.456 | 0.388 | 0.441 | 0.612 | 0.682 | 0.894 | 0.931 |
| ρ/time(×100) ↑ | 0.004 | 0.054 | **0.202** | 0.200 | 0.177 | 0.068 | 0.068 | 0.053 | 0.079 | 0.048 | 0.086 | 0.104 | 0.193 | 0.192 |
| *MNIST* | | | | | | | | | | | | | | |
| Time (s) ↓ | | N.A. | 314 | 306 | 314 | **277** | 293 | 319 | 314 | 326 | 315 | 311 | | N.A. |
| Parameters (m) ↓ | | N.A. | 2.893 | 2.893 | 2.803 | 1.815 | 1.855 | 1.964 | 1.832 | 1.866 | 1.832 | 1.965 | | N.A. |
| Accuracy ↑ | | N.A. | **1.000** | **1.000** | **1.000** | 0.992 | 0.600 | **1.000** | **1.000** | 0.800 | **1.000** | **1.000** | | N.A. |
| FID ↓ | | N.A. | **2.509** | 2.656 | 3.167 | 3.565 | 11.18 | 2.994 | 2.732 | 5.792 | 2.855 | 2.912 | | N.A. |
| FID × time ↓ | | N.A. | **787.8** | 803.5 | 994.4 | 987.5 | 3245 | 955.1 | 857.8 | 1888 | 899.3 | 905.6 | | N.A. |

Several important patterns emerge from these results. The conditioning mechanism choice significantly impacts both training efficiency and accuracy. Our lightweight hypernetwork consistently outperforms alternatives like feature modulation (FiLM) or adaptive normalization approaches (AdaIN, CLN) without major increases in training cost or parameters. Complex approaches like cross-attention prove suboptimal in both accuracy and computational cost. While concatenation offers lowest training time, it achieves worst accuracy on the IAM dataset, confirming our hypothesis that simple conditioning fails when conditions require fundamentally different transport behaviors. Interestingly, adaptive normalization mechanisms like FAN, AdaIn or CLN are good alternatives to the hypernetwork in image generation tasks, which is perhaps unsurprising given their success in different computer vision problems. This also suggests that these alternatives would scale consistently better than cross-attention mechanisms, achieving both relatively expressive transformations while having a reduced training cost.

The continuous variable encoding strategy also proves crucial across both datasets. Positional encoding (our approach) and Fourier features achieve the best accuracy, though Fourier features are less computationally efficient. This holds across both applications despite different continuous variable semantics, suggesting that appropriate processing of raw conditioning information plays a vital role in expressivity.

Our analysis of embedding sharing strategies shows that separating conditioning embeddings for transport and critic networks leads to accuracy gains. This suggests that transport and critic functions utilize conditioning information in fundamentally different ways. This has important implications for conditional transport architecture design. Finally, our pretraining procedure enables higher accuracy with minimal additional training cost, demonstrating that appropriate data-driven initialization improves training dynamics and final results without efficiency penalties.

**Comparisons with previous work** When comparing our framework against the CondOT baselines, the results in 1 show improvements in both expressivity and computational efficiency. On the climate damages dataset, our method outperforms both CondOT Default and CondOT Adapted. In the GSA application, *CondOT Default* fails to capture the underlying sensitivity structures ($\rho = 0.025$), and while the *Adapted* version improves this ($\rho = 0.389$), both fall short of our hypernetwork approach ($\rho = 0.942$). These results

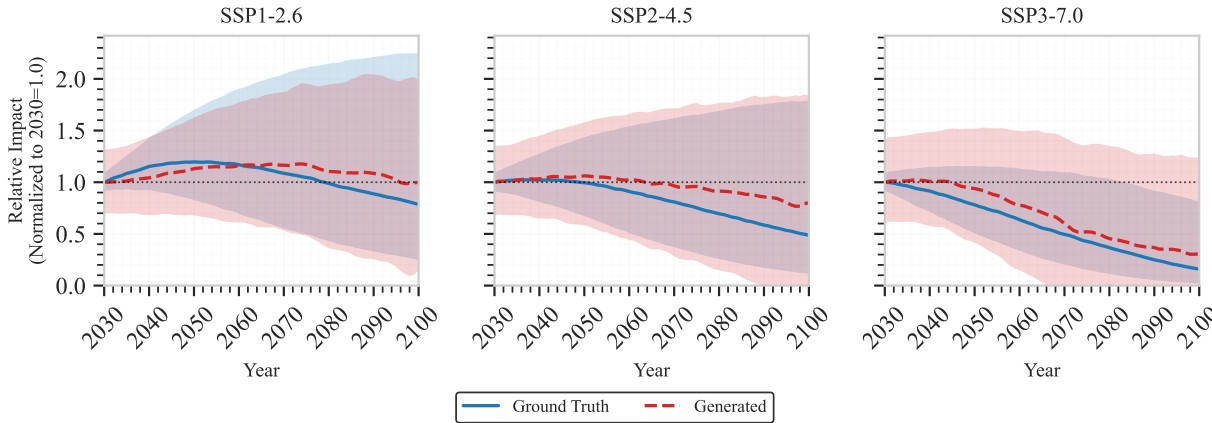

Figure 4: Results of our climate damages model, under different SSP scenarios, on a particular country the dataset. We show the ground truth distribution of damages and samples from our model.

show that both adequate conditioning mechanisms designs and removing the requirements of partial convexity from neural OT architectures allow for more expressive models without increasing overall computational cost.

## 5.2 Application Results

Building on the insights from our ablation studies, we now demonstrate the practical effectiveness of our approach across the scientific and computational applications described in Section 4.

### 5.2.1 Climate Economic Impact Distributions

Figure 4 presents our model's performance on the climate damages dataset, demonstrating its capability to generate realistic distributions across different climate scenarios. For each SSP scenario, we compare ground truth GDP per capita with climate damages distributions against samples from our conditional transport model over the 2030-2100 time horizon.

The results demonstrate our model's effectiveness at capturing both central tendencies and uncertainty (shaded regions represent 90% confidence intervals) specific to each scenario. Our approach successfully learns the distinct temporal evolution patterns across different SSPs: SSP1 shows relatively stable relative impacts with wide uncertainty reflecting the optimistic but uncertain nature of this scenario, while SSP2 exhibits a moderate decline after 2060 consistent with intermediate climate projections. The SSP3 scenario shows the most pronounced downward trend—representing severe climate impacts—which our model captures accurately despite the smaller uncertainty bounds characteristic of this more deterministic high-impact scenario.

### 5.2.2 Global Sensitivity Analysis

Figure 5 presents our validation against traditional simplex-based approaches for computing OT-based sensitivity indices across three input variables in the RICE50+ model. This comparison directly tests our framework's ability to replace computationally intensive traditional methods with efficient neural approximations.

The first three panels show transport costs across different partition values (0 to 1) for each variable, with simplex results in blue and our neural method in red. Our neural transport approach closely tracks the simplex cost patterns across all three variables, effectively capturing the complex sensitivity structure of the underlying model. For the `klogistic` variable (emissions abatement costs), both methods identify similar regions of high sensitivity, with our neural approach maintaining strong correlation ($\rho = 0.942$ from Table 1) with simplex results. The `gamma` variable (inequality aversion) shows more complex sensitivity patterns that our method accurately reproduces, including the pronounced peak near partition value 0.25. For `kw_2`

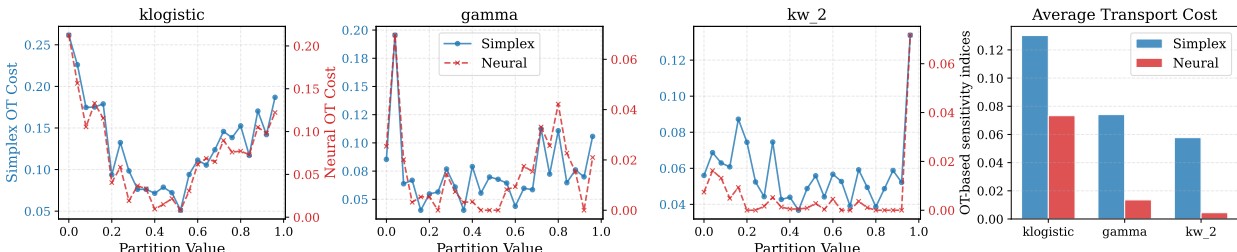

Figure 5: Comparison of simplex and our neural transport across three variables for the IAM dataset. The first three panels show costs across partition values for klogistic, gamma, and kw_2 variables (simplex in blue, neural in red). The rightmost panel shows average costs for each variable.

(climate impacts), both methods detect lower overall sensitivity with consistent patterns across partition values.

The rightmost panel summarizes average transport costs for each variable, revealing that our neural method preserves the relative importance ordering among variables while maintaining comparable absolute cost magnitudes. This preservation of variable ranking is critical for GSA applications where accurate identification of the most influential parameters drives decision-making in policy and scientific contexts.

Crucially, our neural approach achieves this accuracy with dramatically improved scalability—requiring a single trained model rather than solving hundreds of individual OT problems (75 separate optimizations for this three-variable, 25-partition case). This computational advantage becomes increasingly important as model complexity and conditioning dimensionality grow, enabling sensitivity analysis for problems previously considered computationally intractable.

### 5.2.3   Image Generation

Figure 6 presents qualitative results of our ablation study, on MNIST image generation task, using our Conditional Optimal Transport framework. As shown, our baseline configuration (top row) achieves clean digits, significantly outperforming other conditioning methods. Removing the pre-training step or sharing a the conditioning between $T$ and $f$ yield small but noticeable artifacts in the generated digits. Note that MNIST generation is a relatively trivial task for image generative models, and these networks can be trained in approximately 5 minutes on a standard laptop. How these results would scale up to large-scale image generation is outside the scope of this work, as Neural Optimal Transport is not a state-of-the-art formulation for high-resolution controllable image generation, where diffusion models or large-scale generative adversarial networks are currently among the best-performing alternatives.

### 5.3   Computational Analysis and Discussion

The experimental results demonstrate that our conditional neural transport framework successfully addresses the key limitations identified in existing approaches while providing practical advantages for real-world applications.

Our ablation studies reveal that the hypernetwork conditioning mechanism provides the best balance of expressivity and computational efficiency. Unlike feature modulation approaches that can only adjust existing computations, hypernetworks generate fundamentally different transformation parameters for each condition—essential when optimal transport maps must adapt qualitatively across conditioning values. The computational overhead of our lightweight hypernetwork implementation proves minimal compared to the expressivity gains achieved.

The importance of appropriate continuous variable encoding, demonstrated consistently across both climate and GSA applications, highlights an often-overlooked aspect of conditional transport learning. Our sinusoidal positional encoding effectively captures multi-scale variation in continuous conditioning variables, enabling

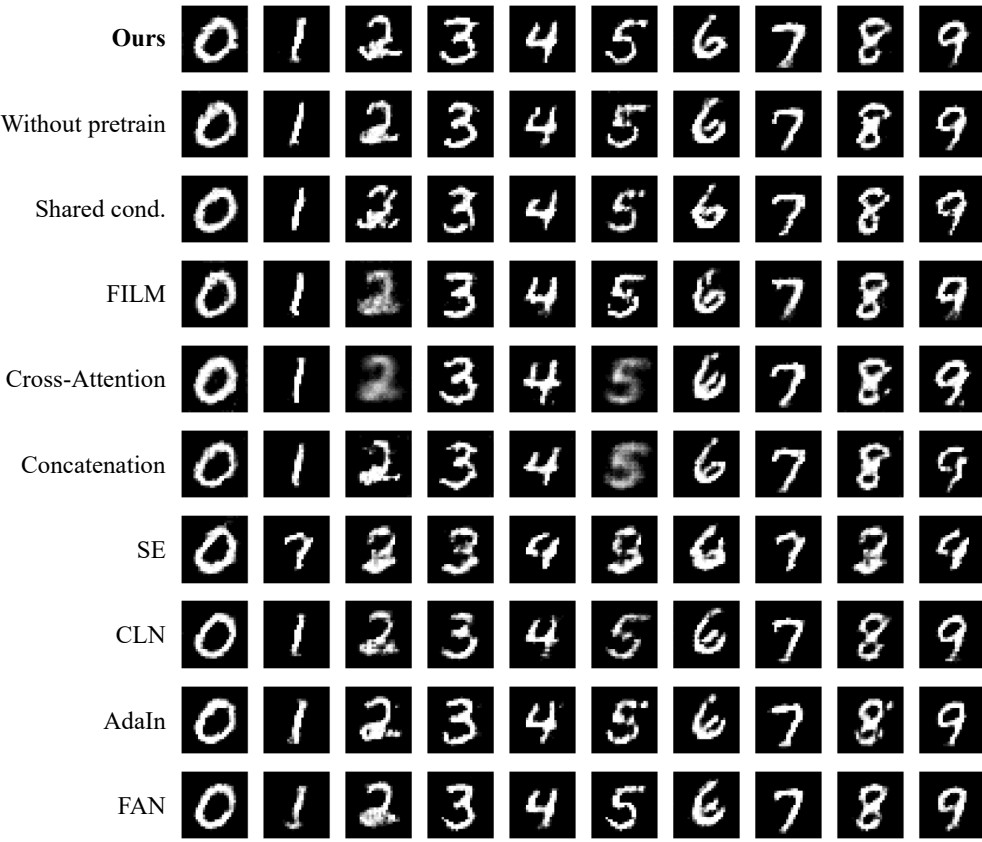

Figure 6: Qualitative ablation study on MNIST digit generation, comparing our method with a hypernetwork (top row) with ablated components, including the lack of pretraining (second row), the used of shared conditioning for $T$ and $f$ (third), and the different conditioning types we tested.

smooth interpolation between conditioning values while maintaining sufficient expressiveness for complex conditional relationships.

The success of our pretraining strategy confirms the importance of initialization in conditional transport learning. By starting transport networks near identity mappings and stabilizing critic functions through multi-objective pretraining, we avoid the optimization pathologies that plague conditional transport training, particularly when handling diverse conditioning values that may require very different optimal mappings.

For practical applications, our approach enables new possibilities in scientific computing and uncertainty quantification. The climate modeling results demonstrate effective emulation of complex multivariate distributions under hybrid conditioning. At the same time, the GSA application shows how computational bottlenecks in sensitivity analysis can be overcome through efficient neural approximation. The preserved statistical properties and correlation with traditional methods indicate that our neural approach can be a reliable replacement for classical techniques in appropriate contexts.

Future work should explore extensions to higher-dimensional conditioning spaces and investigate theoretical guarantees for conditional transport learning. The framework's modular design makes it well-suited for adaptation to emerging applications in generative modeling, domain adaptation, and scientific computing, where conditional optimal transport principles provide natural problem formulations.

# 6 Conclusions

We presented a neural framework for conditional optimal transport that addresses fundamental limitations in existing approaches. Our hypernetwork-based conditioning mechanism addresses three critical bottlenecks: the computational explosion of solving hundreds of individual OT problems in applications like global sensitivity analysis, the architectural constraints of ICNN-based methods that limit expressivity, and the inability of simple conditioning strategies to expressively model transport behaviors across conditions. Our hypernetwork architecture generates transport layer parameters dynamically based on conditioning inputs. This provides the expressiveness of separate networks per condition while maintaining computational efficiency. Our design enables different mappings for different conditions, essential when conditional distributions require distinct optimal transport strategies that feature modulation approaches cannot capture. Importantly, we achieve this without introducing constraints on the neural network architectures (we demonstrate successful cases of MLPs and CNNs), achieving high generality and scope in terms of model design and application domain.

Experiments across scientific computing and image generation tasks show superior performance compared to alternative conditioning approaches. Systematic ablation studies confirm the value of each architectural component. For global sensitivity analysis, our framework replaces hundreds of individual OT problems with a single trained model, achieving high correlation ($\rho = 0.942$) with traditional methods while enabling previously intractable analyses. For climate economic modeling, we efficiently emulate complex multivariate distributions under hybrid conditioning with preserved statistical properties.

**Limitations and Future Work.** Our evaluation focuses primarily on feedforward architectures with residual connections. Recurrent or attention-based architectures could potentially improve performance for sequential conditioning scenarios. While we handle discrete, continuous, and hybrid conditioning effectively, we have not explored complex modalities like CLIP embeddings or pixel-wise semantic labels that could enable more sophisticated generative modeling.

Our hypernetwork approach, though efficient relative to separate networks per condition, still costs more computationally than simple concatenation. More efficient hypernetwork designs or adaptive conditioning mechanisms could reduce this overhead. Theoretical analysis of convergence properties and approximation guarantees would strengthen the foundation for critical applications. Future directions include multi-modal conditioning, connections to diffusion models and normalizing flows, and applications to dynamical systems and causal inference. The modular framework design makes it well-suited for adaptation to emerging generative modeling paradigms and scientific computing applications.

**Broader Impacts.** Our work addresses computational barriers in scientific computing and uncertainty quantification. Making conditional optimal transport computationally tractable enables more comprehensive sensitivity analysis for complex models in climate science, economics, and engineering—potentially improving policy decisions and scientific understanding. The approach also democratizes access to advanced optimal transport techniques for researchers with limited computational resources. For generative modeling, our framework enables more sophisticated conditional generation with principled transport-based approaches, complementing existing diffusion and flow-based methods. This could accelerate progress in controlled content generation, data augmentation, and domain adaptation. However, the ability to generate realistic samples conditioned on specific attributes could be misused for creating misleading synthetic media or other harmful content. Appropriate safeguards include access controls for sensitive applications, combining automated analysis with domain expertise for critical decisions, and providing educational resources to help practitioners understand limitations. The substantial benefits for scientific computing and uncertainty quantification outweigh potential risks when proper oversight is implemented.

# Acknowledgments

Carlos Rodriguez-Pardo, Leonardo Chiani, and Massimo Tavoni acknowledge support from the European Research Council, ERC grant agreement number 101044703 (EUNICE) CUP D87G22000340006.

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
