# OpenReview forum: "Neural Conditional Transport Maps"
_TMLR — Accepted by TMLR_

### Review · Reviewer_yeP3 · 2025-10-31

**Summary Of Contributions:**

Authors introduce neural conditional transport maps: a neural network that transports samples from the source distribution to the target distribution. The transport map is conditioned on an auxiliary variable and optionally stochastic. More specifically, the final approach utilizes hypernetwork that generates the weights of the final layer of the encoder based on the unified conditioning vector.

Strengths:
* The paper is clearly written. I am not very familiar with optimal transport literature but the paper provides a clear introduction to the field that enables understanding of author's contributions.
* The experimental part has sufficient ablation studies and the method is evaluated on 3 different application.

Weaknesses:
* The datasets used in applications seem very toy.

**Additional Comments:**

I am not very familiar with optimal transport literature. Therefore, my confidence is quite low.

**Audience:**

Yes

**Audience Explanation:**

I believe this work would be relevant for audience interested in optimal transport.

**Broader Impact Concerns:**

Broader Impacts section is sufficient for a paper investigating the fundamentals.

**Claims And Evidence:**

Yes

**Claims Explanation:**

The paper provided strong theoretical foundations. The experimental part contains thorough ablation study regarding the architecture of encoder and decoder networks as well as design of conditioning. The method is evaluated in multiple settings, however, the used datasets seem toy.

**Requested Changes:**

None of the following changes is critical but they would strengthen the work:
* Figure 2 provides a general schema for conditional networks. It would be nice to see the diagram of the final approach that utilizes hypernetworks as conditioning. Final approach does not fully match Figure 2 where conditioning is applied on the output of the encoder but hypernetwork generates the weights of the final layer of that encoder which was confusing for me at the beginning.
* Table 1: it is unclear what experiment times and parameters refer to.

Minor writing errors/typos:
* Section 4.1 ends with "-"
* Figure 4: "on a particular country the dataset"

---

### Review · Reviewer_23Ab · 2025-11-10

**Summary Of Contributions:**

The paper introduces a novel conditioning method for Neural Optimal Transport maps, based on a new hypernetwork architecture that generates parameters for optimal transport maps dynamically based on conditioning inputs. The authors test their approach on various datasets, including climate modeling, global sensitivity analysis, and image generation tasks. The paper is well written and provide an extensive introduction and reference to prior work.

**Additional Comments:**

The hypernetwork approach is well-motivated and the experimental validation spans interesting applications. However, the paper would benefit from stronger baselines and more extensive scalability analysis.

**Audience:**

Yes

**Audience Explanation:**

Yes, since it provides at least one novel approach to conditional NOT, leveraging hypernetworks, as well as a comprehensive ablation study on three datasets for other approaches to feature-modulation-type conditioning.

**Claims And Evidence:**

No

**Claims Explanation:**

The paper extends the Neural Optimal Transport framework (Korotin 2023) with a conditioning module based on on hypernetwork. While the architectural design and algorithms are well explained, the experimental results mainly show ablation studies.
- In section 5.1.1 the authors perform an ablation study of architectural and initialization hyperparameter choices from Korotin et al. 2023. It is unclear why such an analysis contributes to the experimental validation of the paper's claims about conditional NOT.
- In section 5.1.2 an extensive ablation study is performed on 3 datasets, extensively evaluating how each architecture and initialization choice impacts the evaluation. However, providing at least one orthogonal method of learning conditional OT maps, such as CondOT would be useful to provide an additional reference for model performance, which does not consist of variations of the NOT framework for Korotin et al. 2023

**Requested Changes:**

- The paper primarily compares against CondOT (Bunne et al., 2022), although only in scope, since it does not report any back to back comparison with ConOT.
- The paper lacks comparison with the more recent work by Wang et al. (2025) that they cite. Given that Wang et al. (2025) also address conditional OT with neural networks, a direct empirical comparison would strengthen the paper's claims. In Wang et al. various approaches are presented. I would suggest the author to extend section 5.1.2 with at least one orthogonal approach to the conditional NOT introduced here, in order to provide a meaningful performance reference to other conditional OT approaches (as the ones presented in Wang et al. 2025)
- The paper claims efficiency advantages but only tests on small-scale problems (20 countries, 58-dimensional outputs, MNIST). Would it be possible for the authors to at least compare a few ablations/baselines with datasets like CIFAR-10 ?
- The paper would benefit from a more concise and clear description of the architecture. The key novelty of the paper, the use of a hypernetwork, is briefly described in section 3.4, first paragraph, but for example, it is not clear how the hypernetwork is trained, and where it fits in the algorithm. 1. Furthermore, a schematic example of how the hypernetwork conditioning compares to standard conditioning (essentially what is explained in section 3.4) would be beneficial and highly increase clarity.
- The choice of Beta(0.95, 0.95) for continuous sampling needs better justification.
- In algorithm 1, the notation L(x, T_θ(x,Z̃_x, c)) is unclear—should this be the cost function k?

---

### Review · Reviewer_5UEZ · 2025-12-05

**Summary Of Contributions:**

The paper extends the Neural Optimal Transport (NOT) framework to the conditional setting, enabling transport maps that depend on both categorical and continuous auxiliary variables. The authors propose a hypernetwork-based conditioning mechanism that generates condition-specific parameters for the transport model, allowing different conditions to induce qualitatively distinct mappings without requiring separate OT solves.
### Key strengths:
- Provides a practical approach for amortizing conditional OT computation, reducing the need for repeated OT solves across many conditions.
- Hypernetwork conditioning offers greater expressiveness than simple concatenation or modulation, and the empirical study systematically compares multiple conditioning architectures.
- Demonstrates applicability to real-world scientific modeling tasks (climate, IAMs) and OT-based sensitivity analysis, where conditional transports are computationally prohibitive with classical OT solvers.
### Key weaknesses:
- The conceptual contribution is incremental, as conditional neural OT was previously introduced (e.g., Supervised Training of Conditional Monge Maps, NeurIPS 2022); the main novelty lies in adopting hypernetworks for conditioning.
- The architectural change, which is generating transport-layer weights via an extra hypernetwork, may be viewed as a modest variation rather than a substantial methodological advance.
- The paper could more clearly articulate the theoretical advantage of hypernetwork conditioning over existing conditional OT parameterizations.

**Audience:**

Yes

**Audience Explanation:**

Researchers working on optimal transport, conditional generative modeling, and scientific applications such as climate or sensitivity analysis may find the empirical insights and practical conditioning strategies of interest, even if the methodological novelty is limited.

**Broader Impact Concerns:**

No major ethical or societal concerns are apparent. The work focuses on methodological advances in optimal transport and conditional modeling, and does not involve sensitive data, human subjects, or deployment-related scenarios.

**Claims And Evidence:**

Yes

**Claims Explanation:**

The paper shows strong experimental results:
- Figure 4 – Climate Damage Model: Shows conditional transport under a continuous scenario parameter
- Figure 5 – IAM Data: Shows conditional transport across categorical scenario partitions, comparing multiple conditioning mechanisms, with the proposed hypernetwork performing best.
- Table 1 – Ablation Study: Summarizes results across tasks, showing that hypernetwork conditioning with pretraining consistently improve performance over alternatives.

**Requested Changes:**

- Provide a visualization or toy example illustrating the claimed limitation of feature-modulation conditioning (e.g., FiLM, concatenation) and how hypernetwork-generated weights enable 'fundamentally different transformations' across conditions. A simple 2D synthetic example where conditions require qualitatively different transport patterns would help substantiate this claim.
- Add discussion comparing the proposed approach to prior neural OT methods, especially conditional neural OT models (e.g., Supervised Training of Conditional Monge Maps, NeurIPS 2022). Please explain specifically:
  - What forms of conditioning these prior methods used
  - Why those mechanisms are insufficient for the scenarios considered here
  - How hypernetwork-based weight generation overcomes those limitations in practice

---

> ### Author Response · Authors · 2025-12-10
> **Response to  5UEZ**
>
> We thank the reviewer for their careful assessment and constructive feedback. We appreciate that they found our method to be practical, applicable to real-world scientific modeling tasks, and sensitivity analysis. We appreciate their acknowledgment of our systematic comparison of conditioning architectures and the relevance to researchers working on optimal transport, conditional generative modeling, and sensitivity analysis. Below we answer specific concerns:
>
> **1. 2D Visualization Demonstrating Conditioning Mechanism Limitations**
>
> We agree this addition would strengthen the paper's intuition. We will include a synthetic 2D example where different conditions require qualitatively different transport behaviors.  We can include this as a new figure in Section 5 or in supplementary materials, complementing our quantitative ablations in Table 1.
>
> **2. Positioning Against Prior Conditional Neural OT Methods**
>
> We will expand Section 2.3 to provide explicit comparison with CondOT (Bunne et al., 2022) and Wang et al. (2025):
> As noted in our response to Reviewer 23Ab, we will also add direct empirical comparisons with CondOT on our benchmark datasets to quantify these architectural differences. We will provide additional discussions regarding what form of conditionings these priors methods use and why these are insufficient for the scenarios we consider in our paper.
>
>
> We thank the reviewer for these constructive suggestions, which will improve both the clarity and completeness of our contribution.

---

### Author Response · Authors · 2025-12-10
**Summary Response to All Reviewers**

We thank all reviewers for their careful reading and constructive feedback. We are encouraged that reviewers found our hypernetwork approach to be "well motivated" (23Ab), our paper "clearly written" (yeP3) or "well written" with "well explained" algorithms (23Ab), our applications to be "interesting" (23Ab) and our work of interest to researchers in optimal transport, conditional generative modeling, and scientific applications (5UEZ). We also find it encouraging that reviewers found our work to be interesting to the TMLR audience (all reviewers) and our claims to be supported by accurate, convincing and clear evidence (5UEZ, yeP3).

We identify three common themes across reviews and summarize our planned revisions below. Detailed responses to each reviewer are provided in individual comments.

**1. Empirical Comparisons with Prior Conditional OT Methods**

Reviewers 5UEZ and 23Ab both request clearer positioning and direct comparisons against prior work, particularly CondOT (Bunne et al., 2022) and Wang et al. (2025). We acknowledge this limitation and commit to:
- Adding direct empirical comparisons with CondOT on our benchmarks
- Expanding Section 2.3 with explicit discussion of what conditioning mechanisms prior methods use and why they are insufficient for our target applications

We estimate approximately 1-2 months to complete these experiments. See our responses to Reviewers 5UEZ and 23Ab for details.

**2. Visualization and Architectural Clarity**

Multiple reviewers requested improved exposition of our hypernetwork conditioning mechanism:
- We will add a 2D synthetic visualization demonstrating why feature modulation approaches are limited in complex transport scenarios, while hypernetwork weight generation succeeds (Reviewer 5UEZ)
- We will include an architectural diagram explicitly comparing standard conditioning with our hypernetwork approach, showing how weights are generated dynamically (Reviewers 23Ab, yeP3)
- We will expand Section 3.5 to clarify training procedures for the hypernetwork

See our responses to Reviewers 5UEZ, 23Ab, and yeP3 for details.

**3. Dataset Scale and Scope**
Reviewer 23Ab suggested CIFAR-10 experiments to demonstrate scalability. We respectfully maintain that our contribution is validated by the scientific computing applications we present, where conditional neural OT provides unique value that other generative approaches cannot address. As we state in Section 5.2.3, neural OT is not competitive with diffusion models or GANs for high-resolution image generation—this is a known limitation of the OT framework, not of our conditioning mechanism. Our efficiency claims concern computational tractability (replacing 75+ separate OT solves with a single model) rather than dimensional scaling. See our response to Reviewer 23Ab for the full discussion.

**Additional Clarifications**

We will also address:
- Justification for Beta(0.95, 0.95) sampling with visualization (Reviewer 23Ab)
- Algorithm 1 notation clarification (Reviewer 23Ab)
- Minor typos (Reviewer yeP3)
- Table 1 caption clarifications (Reviewer yeP3)

We believe these revisions will substantially strengthen the paper and address all major concerns raised. We thank the reviewers again for their time and constructive feedback. We look forward to continued discussion with the reviewers.

---

### Decision · Action_Editor_uYS7 · 2026-02-02

**Recommendation:** Accept with minor revision

**Additional Comments:**

I'm recommending an "Accept with minor revision" since several changes agreed on by authors and reviewers seem fairly central; in particular, a more explicit acknowledgement of prior conditional methods, as well as explicit comparisons to CondOT.

**Audience:**

Yes

**Audience Explanation:**

The novelty of the fundamentals of the proposed approach is limited, but its implementation and results are of interest, since they point to the difficulty of properly training conditional OT mappings.

**Claims And Evidence:**

Yes

**Claims Explanation:**

While the specificity of the paper's claims is limited, experiments (and promised further experiments) create fairly clear evidence.